# Deep Learning-Based Channel Estimation for mmWave Massive MIMO Systems in Mixed-ADC Architecture

**DOI:** 10.3390/s22103938

**Published:** 2022-05-23

**Authors:** Rui Zhang, Weiqiang Tan, Wenliang Nie, Xianda Wu, Ting Liu

**Affiliations:** 1School of Computer Science and Cyber Engineering, Guangzhou University, Guangzhou 510006, China; 2112006270@e.gzhu.edu.cn; 2School of Electronic and Information Engineering, Chongqing Three Gorges University, Chongqing 404000, China; niewenliang@sanxiau.edu.cn; 3School of Electronics and Information Engineering, South China Normal University, Foshan 528000, China; xiandawu@m.scnu.edu.cn; 4School of Artificial Intelligence, Nanjing University of Information Science and Technology, Nanjing 210044, China; liuting@nuist.edu.cn; 5National Mobile Communications Research Laboratory, Southeast University, Nanjing 210096, China

**Keywords:** millimeter-wave, massive MIMO, channel estimation, deep learning, mixed resolution ADC, approximate message passing, compressed sensing

## Abstract

Millimeter-wave (mmWave) massive multiple-input multiple-output (MIMO) systems can significantly reduce the number of radio frequency (RF) chains by using lens antenna arrays, because it is usually the case that the number of RF chains is often much smaller than the number of antennas, so channel estimation becomes very challenging in practical wireless communication. In this paper, we investigated channel estimation for mmWave massive MIMO system with lens antenna array, in which we use a mixed (low/high) resolution analog-to-digital converter (ADC) architecture to trade-off the power consumption and performance of the system. Specifically, most antennas are equipped with low-resolution ADC and the rest of the antennas use high-resolution ADC. By utilizing the sparsity of the mmWave channel, the beamspace channel estimation can be expressed as a sparse signal recovery problem, and the channel can be recovered by the algorithm based on compressed sensing. We compare the traditional channel estimation scheme with the deep learning channel-estimation scheme, which has a better advantage, such as that the estimation scheme based on deep neural network is significantly better than the traditional channel-estimation algorithm.

## 1. Introduction

Millimeter-wave (mmWave) massive multiple-input multiple-output (MIMO) technology can significantly improve data transmission rate with wider bandwidth and higher spectral efficiency, becoming one of the key technologies for the sixth generation (6G) wireless communication [1]. However, digital beamforming and full-resolution ADCs are not suitable in mmWave massive MIMO system, due to the power consumption of ADCs scales being exponential with the number of quantization bits, leading to high hardware cost, power consumption and system complexity. Therefore, the use of current high-speed and high-resolution ADCs (8–12 bits) for each antenna array would become a great burden to the base station (BS) [2,3,4]. Consequently, the use of low-cost and low-resolution ADCs (1–4 bits) is promoted as a potential solution to this problem.

### 1.1. Related Works

In order to reap the maximum benefits of mmWave massive MIMO, the researchers have proposed hybrid precoding, which involves a massive number of antennas attached with a few RF chains [5,6], which effectively improved the energy efficiency of systems. In addition, the authors of [7] utilized the lens antenna array at the BS in the mmWave massive MIMO system, in which signals from different directions can be concentrated on different antennas, and the spatial channel can be converted into a beamspace channel. Thus, the mmWave beamspace channel is sparse and there are only a few main propagation path gains. By selecting a small number of main beams, the number of RF chains required by the massive MIMO system can be significantly reduced [8,9]. Unfortunately, the beam selection requires the BS to obtain the channel state information (CSI), and this is difficult to achieve, especially when the number of RF links is limited. Furthermore, the performance of hybrid precoding systems relies heavily on the precise control of the analog components, and also the selection of the optimal beam will be more difficult if the beam width is small.

In parallel, there is a common solution to replace high-resolution ADCs with low-resolution ADC (1–4 bits), which means the deployment of pure low-resolution ADCs at the BS. However, low-resolution ADCs are prone to severe non-linear distortion, which inevitably causes several problems, including high pilot overhead for channel estimation [10], the system performance loss and signal detection [11]. In order to balance the cost and performance of the system, a mmWave massive MIMO system with a mixed ADC architecture was proposed in [12,13], which replaces the low-resolution ADCs with the partial high-resolution ADCs on the original basis. As reported in [14], channel estimation in a mixed-resolution ADC architecture is easier to process than in a pure low-resolution ADC system. The authors of [15] derived a closed-form approximation of the reachable rate of massive MIMO uplink under Rician fading channels with mixed-ADC. The work in  [16] analyzed the sum-rate performance of the multi-user massive MIMO relaying system equipped with mixed-ADC architecture in the BS. Authors of [17] proposed the channel estimation algorithm for uplinking massive MIMO systems with mixed-ADC architecture. For ease of understanding, the advantages and limitations of the two solutions mentioned above are summarized in Table 1.

Different from using mixed-ADC architecture to reduce cost, another research direction focuses on using the sparsity of the mmWave channel to estimate the channel by using some classical schemes based on compressed sensing (CS) [18,19,20,21]. Specifically, the work in [18] used the orthogonal matching pursuit (OMP) algorithm to detect the dominant entries of multiple channel paths. The authors of [19] proposed a simultaneous weighted orthogonal matching pursuit (SWOMP) channel estimation scheme. Furthermore, the authors of [20] proposed stage wise orthogonal matching pursuit. Using the sparse characteristics of the mmWave channel, the work in [21] designed a compressive sampling matching pursuit, which aims to reduce system complexity. Unfortunately, the estimation accuracy of the greedy algorithm is not ideal in the low signal-to-noise ratio (SNR) range. As a powerful sparse signal recovery algorithm, the approximate message passing (AMP) algorithm was proposed in [22], which can be used to estimate the beamspace channel, but it is difficult to find the optimal solution for the shrinkage parameters of the algorithm.

Deep learning (DL) has achieved great success in speech recognition and image processing. The advantages of DL are expected to bring changes to the communication system. Compared with traditional methods, DL can reveal the internal characteristics of end-to-end collected data or signals, so as to better solve various complicated problems encountered in wireless communication [23,24,25,26,27]. Inspired by the powerful learning ability of deep neural networks (DNNs), some methods based on DL have been applied to channel estimation and achieved good results. For example, the work in [28] applied to channel estimation in the wireless downlink transmission system, which is superior to the traditional scheme in estimation accuracy and energy acquisition. The use of neural networks can help improve channel estimation performance was also demonstrated in [29], where the method was designed by the structure of the minimum mean squared error (MMSE) channel estimator. In addition, the authors of [30] proposed a framework based on DL for direction-of-arrival estimation and the massive MIMO system. In [31], the authors proposed an iterative channel estimation scheme, in which a denoising neural network is used to update the estimated channel in each iteration. The authors in [32] proposed a two-stage massive MIMO channel estimation process based on DL, including pilot-aided and data-aided estimation stages. In [33], a deep neural network based on long short-term memory (LSTM) was introduced to develop a more effective CSI feedback channel compression and recovery method. To improve the estimation accuracy, the authors of [34] developed a learned AMP (LAMP) network for channel estimation. Compared with the original AMP algorithm, the estimation accuracy of both algorithms is improved.

### 1.2. Contributions

In this paper, we investigate channel estimation for mmWave massive MIMO system with lens antenna array, in which we use a mixed (low/high) resolution ADC architecture. To the best of our knowledge, there is no previous work exploring a mixed-ADC architecture of mmWave massive MIMO system and how to estimate CSI based on the DL method. We first study that the mixed-ADC architecture is practically useful because it can provide performance comparable to the ideal high-resolution ADC architecture, while reducing the complexity and power consumption of signal processing. Specifically, the main contributions of this work can be summarized as follows:We study the DL-based channel estimation for a mmWave massive MIMO system with mixed-ADC architecture, where most antennas are equipped with low-resolution ADC and the rest of the antennas use high-resolution ADC. In terms of spectral efficiency, the results showed that the mixed-ADC architecture is practically useful and it can provide performance comparable to the ideal high-resolution ADC architecture.We compare the performance of traditional beamspace channel estimation algorithms and DL-based schemes in mixed-ADC architecture. By applying the sparsity of the mmWave channel, the DL-based channel estimation scheme performs significantly better than the conventional channel estimation algorithm and its complexity outperforms the conventional estimation algorithm.We evaluate the effect of different quantization bits of low-resolution ADC in the mixed-ADC architecture on the channel estimation and the sum rate. The results show that when the number of quantization is about 4 bit, the estimation error with the mixed-ADC architecture is small compared to that with the high-resolution ADC.

Notation: We use boldface letters to denote vectors and capitals to denote matrices. For matrix A, $A_{ij}$ is the (i,j) th entry of A. $A^{T},A^{H}$ represent A’s transpose, complex conjugate. Moreover, a circularly symmetric complex Gaussian random vector with zero mean and covariance matrix R is denoted v∼CN(0,R), $I_{n}$ is the identity matrix of size *n*, ⊗ is the Kronecker product and ${∥\cdot ∥}_{2}$ is the Euclidean norm.

## 2. System Model

We consider a time division duplex (TDD) mmWave massive MIMO system, in which the BS is equipped with *N* antennas and $N_{RF}$ RF chains, serving *K* single antenna users simultaneously In this section, we first introduce the traditional mmWave massive MIMO channel and the space channel after adding the lens antenna array. In the second section, we introduce the mixed resolution ADC architecture. Finally, we express the beamspace channel estimation problem as the sparse signal recovery problem.

### 2.1. Beamspace Channel Model

We start with the traditional massive MIMO mmWave channel, and in this work, we adopt the Saleh–Valenzuela channel model, which is widely used in the frequency domain. The channel vector with the size of N×1 between the *k*th (k=1,2,…,K) user and the BS can be expressed as
(1)$$h_{k}=\sqrt{\frac{N}{L_{k}}}\sum_{l=1}^{L_{k}}\beta_{k,l}a\left(\phi_{k,l},\theta_{k,l}\right)=\sqrt{\frac{N}{L_{k}}}\sum_{l=1}^{L_{k}}c_{k,l},$$
where *N* is the total number of antennas, $L_{k}$ is the number of resolvable paths and $\beta_{k,l}$, $\phi_{k,l}$ and $\theta_{k,l}$ are the complex gain, azimuth and elevation on the *l*th path, respectively. $a\left(\phi_{k,l},\theta_{k,l}\right)$ is the N×1 array steering vector, which depends on the array geometry, and $c_{k,l}=\beta_{k,l}a\left(\phi_{k,l},\theta_{k,l}\right)$ denotes the *l*th path component. For a typical uniform planar arrays (UPAs) with $N_{1}\times N_{2}\;\left(N=N_{1}\times N_{2}\right)$ antennas, the array steering vector can be expressed as
(2)$$\begin{array}{c} a\left(\phi ,\theta \right)\;\;=\;\;\frac{1}{\sqrt{N}}\;\;\left[e^{-j2\pi d\sin\left(\phi \right)\sin\left(\theta \right)n_{1}/\lambda }\right]\;\;⊗\;\;\left[e^{-j2\pi d\cos\left(\theta \right)n_{2}/\lambda }\right], \end{array}$$
where $n_{1}={\left[0,1,\cdots ,N_{1}-1\right]}^{T}$ and $n_{2}=$${\left[0,1,\cdots ,N_{2}-1\right]}^{T},\lambda$ is the wavelength of the carrier, and *d* is the antenna spacing usually satisfying d=λ/2 in mmWave communications. Then, we can define $\psi_{a}=d\sin\left(\phi \right)\sin\left(\theta \right)/\lambda$ and $\psi_{e}=d\cos\left(\theta \right)/\lambda$, respectively, as the spatial angles for UPAs configuration [35].

Traditional channels in the spatial domain can be converted to beam spatial channels by using a lens antenna array. In fact, the lens antenna array plays the role of the spatial discrete Fourier transform (DFT) matrix U of size N×N. For UPA configuration, the matrix U can be expressed as
(3)$$\begin{array}{cc} \; & U={\left[a\left(\psi_{a,1},\psi_{e,1}\right),\cdots ,a\left(\psi_{a,1},\psi_{e,N_{2}}\right),\cdots ,a\left(\psi_{a,N_{1}},\psi_{e,1}\right),\cdots ,a\left(\psi_{a,N_{1}},\psi_{e,N_{2}}\right)\right]}^{H}, \end{array}$$
where $\psi_{a,n}=\frac{1}{N_{1}}\left(n-\frac{N_{1}+1}{2}\right)$ for $n=1,2,\cdots ,N_{1}$ and $\psi_{e,n}=\frac{1}{N_{2}}\left(n-\frac{N_{2}+1}{2}\right)$ for $n=1,2,\cdots ,N_{2}$ are the spatial direction corresponding to the azimuth and elevation predefined by the lens antenna array, respectively. Therefore, the beamspace channel vector ${\bar{h}}_{k}$ with a size of N×1 between the *k*th user and BS with *N* antennas can be written as
(4)$${\bar{h}}_{k}={\mathrm{Uh}}_{k}=\sqrt{\frac{N}{L}}\sum_{l=1}^{L}{\bar{c}}_{k,l}$$
where ${\bar{c}}_{k,l}={\mathrm{Uc}}_{k,l}$ represents the *l*th path component in the beamspace channel.

### 2.2. Mixed Resolution ADC Architecture

In this section, we consider a mixed resolution ADC architecture on the part of the lens antenna array. We divide the antenna into two parts, where the $N_{0}=\sigma N$ antennas are connected to the high-resolution ADC, and the $N_{1}=\left(1-\sigma \right)N$ antenna is connected to lower ADC. In order to facilitate calculation, the antenna number $N_{0}$ and $N_{1}$ in the simulation are integers, and the coefficient σ is also limited to a certain rational number. In this experiment, σ is set to 0.25, so we can redivide the channel matrix and rewrite the above Formula (Equation 4) as
(5)$${\tilde{h}}_{k}=\left[\begin{array}{c} {\bar{h}}_{k,0}\\ {\bar{h}}_{k,1} \end{array}\right]=U\left[\begin{array}{c} h_{k,0}\\ h_{k,1} \end{array}\right],$$
where ${\bar{h}}_{k,0}$ represents the channel matrix associated with $N_{0}$ antennas connected to a high resolution ADC, and ${\bar{h}}_{k,1}$ represents the channel matrix associated with $N_{1}$ antennas connected to a low resolution ADC.

Due to the reciprocity of the TDD channel, after the pilot sequence is transmitted to BS, the user can obtain the downlink channel through the estimated uplink channel. This paper adopts the widely used orthogonal pilot transmission strategy, and the channel estimation of each user is independent, so the subscript *k* in Formula (Equation 4) can be omitted.

Assuming that all users transmit known pilot sequences through instance *Q* for channel estimation, the measured signal $y_{q}$ with a size of N×1 passing through the RF chains at the *q*th instant can be expressed as
(6)$$y_{q}=A_{q}\tilde{h}s_{q}+{\bar{n}}_{q},\;q=1,2,\cdots ,Q,$$
where $A_{q}$ is the $N_{\mathrm{RF}}\times N$ beam-selection network, $s_{q}$ is the pilot transmitted symbol at the *q*th instant, ${\bar{n}}_{q}=A_{q}n_{q}$ is the effective noise vector, where $n_{q}∼CN\left(0,\sigma_{n}^{2}I_{N}\right)$ is the N×1 noise vector with $\sigma_{n}^{2}$ representing the noise power.

After the pilot transmission of *Q* instances, we can obtain the $QN_{\mathrm{RF}}\times 1$ overall measurement vector y by assuming $s_{q}=1$ for q=1,2,⋯,Q as
(7)$$y={\left[y_{1},y_{2},\ldots ,y_{Q}\right]}^{T}=A\tilde{h}+n,$$
where $A={\left[A_{1}^{T},A_{2}^{T},\cdots ,A_{Q}^{T}\right]}^{T}$ is the $QN_{\mathrm{RF}}\times N$ overall combining matrix, and n=${\left[{\bar{n}}_{1}^{T},{\bar{n}}_{2}^{T},\cdots ,{\bar{n}}_{Q}^{T}\right]}^{T}$ is the effective noise vector for *Q* instants.

### 2.3. Problem Formulation

According to Formula (Equation 7), we can now recover channel $\tilde{h}$ from y and A. Due to the limited scattering at mmWave frequency, there are only a few propagation paths, and the beam space channel $\tilde{h}$ is approximately sparse. This problem can be solved by the sparse signal recovery algorithm in compressed sensing (CS), in which matrix A in Formula (Equation 7) can be regarded as the sensing matrix in CS.
(8)$$\min∥\tilde{h}{∥}_{0},\;\;s.t.\;{∥y-A\tilde{h}∥}_{2}\leq \varepsilon ,$$
where $∥\tilde{h}{∥}_{0}$ is the number of non-zero elements of $\tilde{h},\varepsilon$ is the error tolerance parameter.

Because the $l_{0}$-norm minimization problem is a NP-hard problem in practice, so in many cases, the $l_{0}$-norm optimization problem will be converted to a higher-dimension norm problem, such as replacing $l_{0}$-norm with $l_{1}$-norm for the convex optimization problem. At present, some traditional greedy algorithms are commonly used to solve this problem; for example, the OMP [18] and compressive sampling matching pursuit (CoSaMP) [21]. However, using these greedy algorithms it is difficult to find the global optimal solution and the estimation accuracy is not ideal.

## 3. Traditional Channel Estmation Algorithems

### 3.1. Compressed Sensing

Compressed sensing is a new technology for finding sparse solutions of underdetermined linear systems. By using the characteristics of signal sparsity, compared with Nyquist theory, the technology can restore the original signal to be recognized from fewer measured values. The theory mainly includes three parts: signal-sparse representation, reconstruction conditions and signal recovery algorithm. Suppose a signal $x\in R^{N\times 1}$ can be sparsely represented by an orthonormal basis $\Psi \in R^{N\times N}$, i.e,
(9)x=Ψs,
where s is a sparse column vector, the sparse coefficient of x is *K* and *K* is far less than *N*. Consider a linear measurement process; x is represented by an orthonormal basis Ψ and a coefficient vector s, and the measured value $y\in R^{M\times 1}\left(M<<N\right)$ can be written as
(10)y=Φx=ΦΨs=As,
where Φ represents the measurement matrix, and A=ΦΨ is the sensing matrix.

In general, taking the sparse of the reconstructed signal in a certain transform domain as a priori information, the original signal is observed with the measurement matrix, and the complete measured signal is reconstructed from the observed value combined with the reconstruction algorithm.

### 3.2. OMP (Orthogonal Matching Pursuit)

Orthogonal Matching Pursuit is one of the classic algorithms in the field of compressed sensing. It is the basis of many commonly used efficient algorithms. This algorithm has the characteristics of simplicity and efficiency. The essence of the OMP algorithm is to select the columns in the sensor matrix by greedy iteration so that the selected column is most related to the current redundant vector in the process of each iteration, subtract the relevant part from the original signal vector, and iterate repeatedly until the number of iterations reaches the sparsity *K*, and then stop the iteration.

We define $A\in C^{M\times N}$ as a sensing matrix and $a_{i}$ is the ith column of A, observation vector $y\in C^{M\times 1}$ and sparsity *K*, the initialization residual $r_{0}=y$. The initial iterated index collection $\Lambda_{0}=⌀$, and initial value of iteration k=1. To solve the first problem, we attempt to look for an iterative index as follows:(11)$$\lambda_{k}=\arg\max_{i=1,\cdots ,N}\left|\langle r_{k-1},a_{i}\rangle \right|.$$

Adds the index of the most relevant dictionary element found to the index set, while the set of reconstructed atoms in the sensing matrix is updated:(12)$$\Lambda_{k}=\Lambda_{k-1}\cup \left\{\lambda_{k}\right\},$$
(13)$$A_{\Lambda_{k}}=A_{\Lambda_{k-1}}\cup a_{\Lambda_{k}},$$

Update the residual:(14)$$r_{k}=y-A_{\Lambda_{k}}{\left(A_{\Lambda_{k}}^{T}A_{\Lambda_{k}}\right)}^{-1}A_{\Lambda_{k}}^{T}y=y-A_{\Lambda_{k}}s_{k}.$$

After executing *K* cycles, the reconstructed sparse coefficient $s_{k}$ can be obtained. For more specific OMP algorithm flow, please refer to Algorithm 1.
**Algorithm 1** OMP Algorithm in mmWave Channels**Require:** 
   sensing matrix A, measurement vector y, and the threshold δ   1:$\Lambda_{0}=⌀,r_{0}=y,r_{-1}=0$, and k=1   2:$\mathrm{while}{∥r_{k-1}-r_{k-2}∥}_{2}^{2}>\delta$ and $k\leq k_{\max\;}\mathrm{do}$   3:$\lambda_{k}=\underset{i=1,\cdots ,N}{\mathrm{argmax}}\left|\langle A^{H}\left(i\right),r_{k-1}\rangle \right|$   4:$\Lambda_{k}=\Lambda_{k-1}\cup \left\{\lambda_{k}\right\}$   5:$A_{\Lambda_{k}}=A_{\Lambda_{k-1}}\cup a_{\Lambda_{k}}$   6:$s_{k}={\left(A_{\Lambda_{k}}^{T}A_{\Lambda_{k}}\right)}^{-1}A_{\Lambda_{k}}^{T}y$   7:$r_{k}=y-A_{\Lambda_{k}}s_{k}$   8:k=k+1   9:endwhile 10:**return**$s_{k}$

### 3.3. AMP (Approximate Message Passing)

The approximate message passing (AMP) algorithm is an iterative compressed sensing approach based on a probability graph to predict the next iteration through state evolution and to de-noise through soft threshold iteration. In this section, we will describe how the AMP algorithm estimates the beamspace channel, as shown in Algorithm 2.
**Algorithm 2** AMP Algorithm in mmWave Channels**Require:** 
   sensing matrix A, measurement vector y, and the number of iterations *T*   1:$\mathrm{Initialize}\;\;{\hat{h}}_{0}=0,v_{-1}=0$, and $b_{0}=0$   2:fort=0,1,⋯,T−1do   3:$v_{t}=y-A{\hat{h}}_{t}+b_{t}v_{t-1}$   4:$r_{t}={\hat{h}}_{t}+A^{T}v_{t}$   5:$\lambda_{t}=\frac{1}{\sqrt{M}}{∥v_{t}∥}_{2}$   6:${\hat{h}}_{t+1}=\eta_{st}\left(r_{t};\alpha ,\lambda_{t}\right)$   7:$b_{t+1}=\frac{1}{M}\sum_{i=1}^{N}\frac{\partial \eta_{\mathrm{st}}\left(r_{t};\alpha ,\lambda_{t}\right)}{\partial r_{t,i}}$   8:endfor   9:**return**$\hat{h}={\hat{h}}_{T}.$

In Algorithm 2, α in Step 6 is a tuning parameter and usually takes a fixed value during iteration, the term $b_{t}v_{t-1}$ in Step 3 is called Onsager Correction, which is introduced into the AMP algorithm to accelerate the convergence. The key step of the AMP algorithm is step 6, in which the estimated channel ${\hat{h}}_{t+1}$ is obtained by soft threshold shrinking function $\eta_{\mathrm{st}}$ during the $t^{th}$ iteration. The shrinkage function $\eta_{\mathrm{st}}$ is non-linear element-wise operation, due to the sparsity of the channel, the channel vector ${\hat{h}}_{t+1}$ updated in each iteration will be more sparse. For the *i* th element $r_{t,i}=\left|r_{t,i}\right|e^{j\omega_{t,i}}\left(i=1,2,\cdots ,N\right)$ of input vector $r_{t}$, we have
(15)$$\begin{array}{cc} {\left[\eta_{\mathrm{st}}\left(r_{t};\alpha ,\lambda_{t}\right)\right]}_{i} & =\eta_{\mathrm{st}}\left(\left|r_{t,i}\right|e^{j\omega_{t,i}};\alpha ,\lambda_{t}\right)\\ \; & =\max\left(\left|r_{t,i}\right|-\alpha \lambda_{t},0\right)e^{j\omega_{t,i}}, \end{array}$$
where $\omega_{t,i}$ is the phase of complex-valued element $r_{t,i}$, α is the fixed parameter in the *T* iteration, and $\lambda_{t}$ is updated with iteration process in Step 5. In addition, *b* is obtained by calculating the element-wise derivatives of the shrinkage function at the input vector *r* in Step 7.

Although the AMP algorithm can effectively deal with massive sparse signal problems, and performs well in many traditional channel-estimation algorithm schemes, for sparse beamspace channel estimation, many problems still exist, such as that the AMP algorithm has a high requirement for i.i.d. sub-Gaussian matrix A. Otherwise, the algorithm itself is prone to divergence. There are two key problems that restrict the performance of the AMP algorithm: (1) The shrinkage parameters in the AMP algorithm usually take the same value in the whole iterative process; (2) the general AMP algorithm cannot make full use of the prior distribution of beamspace channels.

## 4. AI Channel Estmation Algorithems

### 4.1. Deep Learning

Deep learning is a kind of representation learning method based on data in machine learning. Its basis is the neural network. In DL, training data ${\left\{\left(y^{d},x^{d}\right)\right\}}_{d=1}^{D}$ consisting of feature and label pairs are used to train the parameters of the deep neural network, which aims to accurately predict the unknown label x associated with the newly acquired feature y. Depth networks accept y and process it in many layers, each of which usually consists of a linear transformation followed by a simple non-linear transformation. Unlike traditional shallow learning, in-depth learning emphasizes the depth of the model structure, usually with five, six or even ten layers of hidden nodes, and transforms the feature representation of samples in the original space into a new feature space through layer-by-layer feature transformation, which makes classification or prediction easier.

In general, the label space is discrete; for example, y is an audio or a picture; x is a class of cat, dog, or some other type. However, for the sparse linear problem in beam space, label x is continuous, but before that, some authors have demonstrated that a well-constructed deep neural network can predict such tags accurately.

### 4.2. LAMP Network

Recently, the authors of [34] proposed a LAMP network scheme based on the classical AMP algorithm. The results showed that the contraction parameters of the AMP algorithm can only take the same value during the iterative process. Therefore, it essence is to map each iteration of the AMP algorithm to each layer of the LAMP network and optimize the non-linear parameters α in each iteration.

Figure 1 shows the network structure of the LAMP scheme with a total of *T* layers. Specifically, the LAMP network processes signals in the same way as the AMP algorithm, where the input of *t*th layer is $y,\;{\hat{h}}_{t}$ and $v_{t}$. It is worth mentioning that y is the measurement signal, and both ${\hat{h}}_{t}$ and $v_{t}$ are (t−1)th layer outputs, so the processing of signals in *t*th layer can be summarized as follows:(16)$${\hat{h}}_{t+1}=\eta_{st}\left(r_{t};\alpha_{t},\lambda_{t}\right),$$
(17)$$v_{t+1}=y-A{\hat{h}}_{t+1}+b_{t+1}v_{t},$$
where
(18)$$\lambda_{t}=\frac{1}{\sqrt{M}}{∥v_{t}∥}_{2},$$
(19)$$r_{t}={\hat{h}}_{t}+B^{T}v_{t},$$
(20)$$b_{t+1}=\frac{1}{M}\sum_{i=1}^{N}\frac{\partial \eta_{\mathrm{st}}\left(r_{t};\alpha ,\lambda_{t}\right)}{\partial r_{t,i}}.$$

From the above Formulas (Equation 16) and (Equation 19), we see the LAMP network involved in the learnable parameters compared with Algorithm 2. In the *t*th iteration, operations involving $\left(\alpha ,\;A^{T}\right)$ are replaced by $\left(\alpha_{t},\;B^{T}\right)$, and the shrinkage function $\eta_{st}$ of the AMP algorithm plays a role in the non-linear activation function in the conventional DNN. It is worth noting that selecting $A^{T}$ in the AMP algorithm is only for the convenience of formula derivation. If enough training data are given, the LAMP network can use DNN’s powerful learning ability to find better shrinking parameters. Thus, the performance of the original AMP algorithm can be further improved by optimizing the linear transformation coefficient $B^{T}$ and non-linear shrinking parameter $\alpha_{t}$.

### 4.3. GM-LAMP Network

The LAMP network solves the problem of taking fixed shrinkage parameters in the AMP algorithm, but it does not make good use of prior information on the beamspace channel. The GM-LAMP network proposed by [36] solves this problem. The GM-LAMP network derives a new shrinking function by considering the Gaussian mixing distribution of the elements of the beamspace channel. Before that, we first introduce the expression of the probability density function of the element $\tilde{h}$ in the beamspace channel:(21)$$p\left(\tilde{h};\theta \right)=\sum_{k=0}^{N_{c}-1}p_{k}CN\left(\tilde{h};\mu_{k},\sigma_{k}^{2}\right)$$
where $\theta =\left\{p_{0},\cdots ,p_{N_{c}-1},\mu_{0},\cdots ,\mu_{N_{c}-1},\sigma_{0}^{2},\cdots ,\sigma_{N_{c}-1}^{2}\right\}$ is a set of all the parameters, $p_{k}$ is the probability of the kth Gaussian component and $N_{c}$ is the number of Gaussian components in the Gaussian mixture distribution. $\mu_{k}$ and $\sigma_{k}^{2}$ represent the mean and variance of the kth Gaussian component, respectively. $CN\left(\tilde{h};\mu_{k},\sigma_{k}^{2}\right)=\frac{1}{\pi \sigma_{k}^{2}}e^{-\frac{{\left(\tilde{h}-\mu_{k}\right)}^{*}\left(\tilde{h}-\mu_{k}\right)}{\sigma_{k}^{2}}}$ denotes the probability density function of the *k* th Gaussian component.

It is interesting that when the mean and variance of Gaussian components are zero, the probability density function of Gaussian distribution can be rewritten as
$$\mathrm{CN}\left(\tilde{h};0,0\right)=\delta \left(\tilde{h}\right),$$
where the $\delta (\tilde{h})$ is the Dirac delta function, which means that the variable $\tilde{h}$ will be exactly zero. Therefore, the sparsity of beam space channel can be described as a special case by using Gaussian mixture distribution. Finally, the Gaussian Mixed Shrinkage Function considering the prior distribution of the beamspace channel can be written as
(22)$$\eta_{\mathrm{gm}}\left(r;\theta ,\sigma^{2}\right)=\frac{\sum_{k=0}^{N_{c}-1}p_{k}{\tilde{\mu }}_{k}\left(r\right)CN\left(r;\mu_{k},\sigma^{2}+\sigma_{k}^{2}\right)}{\sum_{k=0}^{N_{c}-1}p_{k}CN\left(r;\mu_{k},\sigma^{2}+\sigma_{k}^{2}\right)},$$
where a set of all distribution parameters θ can also be called as the shrinkage parameters.

GM-LAMP network is still based on AMP algorithm, and it has T uniform layers, in which the input and output of each layer are the same as that of LAMP network, and the difference from LAMP network is that its soft threshold shrinkage function is replaced by the Gaussian mixture shrinkage function. The channel estimation for the *t*th layer can be written as
(23)$$r_{t}={\hat{h}}_{t}+B^{T}v_{t},$$
(24)$${\hat{h}}_{t+1}=\eta_{\mathrm{gm}}\left(r_{t};\theta_{t},\sigma^{2}\right),$$
where the linear transformation coefficient is $B^{T}$ and the non-linear shrinkage parameter $\theta_{t}$ is the variable that can be optimized in the training stage.

The GM-LAMP network is mainly divided into two stages: offline training and online estimation. During the offline training phase, a large amount of known training data are provided to optimize the overall trainable variable by minimizing the loss function. In the online estimation phase, the new measurement data can be input into the trained GM-LAMP network and the corresponding channel estimates can be obtained directly.

In the offline training stage, supervised learning is adopted to train the GM-LAMP network, and the training dataset can be expressed as ${\left\{y^{d},{\tilde{h}}^{d}\right\}}_{d=1}^{D}$, where $y^{d}$ is the input of the GM-LAMP network, ${\tilde{h}}^{d}$ is the corresponding label and *D* represents the number of the training data.

In order to avoid over-fitting, a layer-by-layer training method is adopted. Specifically, the whole training process can be divided into *T* training subroutines according to 0,1,⋯,T−1 sequence. For the *t*th training subprocess, its objective is to optimize trainable variable $\Omega_{t}=$${\left\{B_{i},\theta_{i}\right\}}_{i=0}^{t}$ of the $i=0,\cdots ,i=t^{th}$ layer. In this simulation, we refer to the GM-LAMP algorithm that was first proposed in the literature [36]. In the model training stage, the huber loss is used to define two loss functions of linear transformation coefficient $B_{t}$ and non-linear shrinkage parameter $\theta_{t}$:(25)$$\;L_{t}^{\mathrm{linear}}\left(\Omega_{t}\right)\;=\;\;\left\{\begin{array}{cc} \frac{1}{2}{\left(r_{t}^{d}\left(y^{d},\Omega_{t}\right)-{\tilde{h}}^{d}\right)}^{2} & V_{1}\leq \delta \\ \delta |r_{t}^{d}\left(y^{d},\Omega_{t}\right)-{\tilde{h}}^{d}|-\frac{1}{2}\delta^{2} & \mathrm{otherwise} \end{array}\right.,$$
(26)$$L_{t}^{\mathrm{non}-\mathrm{linear}}\left(\Omega_{t}\right)\;\;=\;\;\left\{\begin{array}{cc} \frac{1}{2}{\left({\hat{h}}_{t+1}^{d}\left(y^{d},\Omega_{t}\right)-{\tilde{h}}^{d}\right)}^{2} & V_{2}\leq \delta \\ \delta |{\hat{h}}_{t+1}^{d}\left(y^{d},\Omega_{t}\right)-{\tilde{h}}^{d}|-\frac{1}{2}\delta^{2} & \mathrm{otherwise} \end{array}\right.,$$
where $V_{1}=\left|r_{t}^{d}\left(y^{d},\Omega_{t}\right)-{\tilde{h}}^{d}\right|$, $V_{2}=\left|{\hat{h}}_{t+1}^{d}\left(y^{d},\Omega_{t}\right)-{\tilde{h}}^{d}\right|$, δ is a hyperparameter, $r_{t}^{d}$ is the output of linear transformation operation in Formula (Equation 23), and ${\hat{h}}_{t+1}^{d}$ is the output of non-linear contraction operation in Formula (Equation 24). Based on these two loss functions, the training sub-process at the *t*th layer can be divided into linear training to minimize $L_{t}^{\mathrm{linear}\;}$ and non-linear training to minimize $L_{t}^{\mathrm{non}-\mathrm{linear}\;}$.

Algorithm 3 represents the specific layer-by-layer training method of the GM-LAMP network that was provided in [36]. To avoid the trapped-in local optimization caused by over-fitting, we firstly adopt the separate optimization method in step 2, and then jointly optimize the method in step 3 and 4, which are set to $B_{0}$ and $\theta_{0}$, respectively. Then the training is carried out in sequence from the 1th layer to the (T−1)th layer. For the training sub-process of the *t*th layer, the training variable is set to the value of the training variable of the (t−1)th layer before training. Steps 7–8 indicate that the linear transformation coefficient $B_{t}$ is optimized separately, and then $B_{t}$ and $\Omega_{t}$ are optimized jointly. Similarly, steps 9–10 indicate that the non-linear shrinkage parameter $\theta_{t}$ is optimized separately first, and then $B_{t}$, $\Omega_{t}$ and $\theta_{t}$ are optimized jointly.
**Algorithm 3** Layer-by-Layer Training Method   1:$\mathrm{Initialize}\;\;B_{0}=A^{T}$ and $\theta_{0}$   2:Learn $B_{0}$ by minimizing $L_{t}^{\mathrm{linear}\;}$   3:Learn $\theta_{0}$ with fixed $B_{0}$ by minimizing $L_{t}^{\mathrm{non}-\mathrm{linear}\;}$   4:Relearn $\Omega_{0}=\left\{B_{0},\theta_{0}\right\}$ to minimize $L_{0}^{\mathrm{non}-\mathrm{linear}\;}$   5:fort=1,2,⋯,T−1do   6:$B_{t}=B_{t-1},\theta_{t}=\theta_{t-1}$   7:Learn $B_{t}$ with fixed $\Omega_{t-1}$ by minimizing $L_{t}^{\mathrm{linear}\;}$   8:Relearn $\left\{{B_{t},\Omega }_{t-1}\right\}$ by minimizing $L_{t}^{\mathrm{linear}\;}$   9:Learn $\theta_{t}$ with fixed $\left\{{B_{t},\Omega }_{t-1}\right\}$ by minimizing $L_{t}^{\mathrm{non}-\mathrm{linear}\;}$ 10:Relearn $\left\{{B_{t},\Omega }_{t-1},\theta_{t}\right\}$ by minimizing $L_{t}^{\mathrm{non}-\mathrm{linear}\;}$ 11:endfor 12:**return**$\Omega_{T-1}$

A trained GM-LAMP network can be obtained after optimizing the overall trainable variables $\Omega_{T-1}$ of *T* layers. In the online estimation phase, the corresponding estimates can be directly generated by inputting new measurement signals into the trained GM-LAMP network.

### 4.4. Complexity Analysis

In this subsection, we perform the complexity analysis of the channel estimation algorithms used in the simulation process. Considering the OMP algorithm, when the sparsity of the channel vector is set to *K*, the computational complexity in the atom selection step is about O(KMN), and the computational complexity of the *k*-th iteration is at least $O(K^{3})$. Therefore, we ignore trivial operations; the computational complexity can be calculated as $O\left(KMN\right)+O\left(K^{3}M\right)$. In addition, since the LAMP network and GM-LAMP network are constructed that both are based on the AMP algorithm, the computational complexity of the AMP algorithm, LAMP network and GM-LAMP network is roughly the same values that equals to O(TMN).

## 5. Numerical Results

In this section, we compare the performance of traditional beamspace channel estimation algorithms and DL-based schemes in mixed-ADC architecture. Specifically, DL-based schemes include the LAMP network and the GM-LAMP network. This experiment provides a widely used Saleh–Valenzuela channel model.

### 5.1. Parameter Setting

In our simulations, we consider that the BS is deployed a lens antenna array, where the numbers of antennas and RF chains are set to N=256 and $N_{RF}=16$, individually. The number of single-antenna users is set to K=16, and the number of measurements is set to M=128. For the the Saleh–Valenzuela channel model, we set the same channel parameters for each user *k*, where the number of channel paths $L_{k}=3$, the complex gain on the *l*th path satisfies $\beta_{k,l}∼CN\left(0,1\right)$ for l=1,2,3, and the range of azimuth $\phi_{k,l}$ and elevation $\theta_{k,l}$ on each path is between −π/2 and π/2. In order to train and test the LAMP network and the GM-LAMP network, we generate 80,000, 2000 and 2000 samples as the training, the validation and the testing set based on the above setup, respectively. Then the number of training layers is set as T=8 for both network schemes, where the number of nodes in each layer depends on the number of measurements *M* and the number of dimensions *N* of the beamspace channel. Finally, we use the normalized mean square error (NMSE) to quantify the accuracy of channel estimation for each user, which is mathematically defined as
(27)$$\mathrm{NMSE}=10\log_{10}\left(E\left[\frac{∥\hat{h}-\tilde{h}{∥}^{2}}{∥\tilde{h}{∥}^{2}}\right]\right),$$
where $\hat{h}$ is the recovered channel matrix using the channel estimation algorithms.

### 5.2. Simulation Results on the Saleh–Valenzuela Channel Model

Figure 2 shows the NMSE performance comparison of various algorithms under the considered UPA and mixed-resolution ADC quantization architecture. It can be seen from the figure that the performance of the traditional two algorithms is poor, and in general, the performance of the AMP algorithm is better than the OMP algorithm. For the DL-based LAMP network and GM-LAMP network, the performance of the scheme has been greatly improved. In addition, considering the prior distribution of beamspace channels, the GM-LAMP network has better channel estimation accuracy than the LAMP network.

Figure 3 shows a comparison of the complexity of the above four kinds of channel estimation schemes. We can find that the traditional OMP algorithm requires more complex multipliers than other schemes, because the OMP algorithm is a kind of greed-tracking algorithm, the number of complex multiplications increases exponentially with the number of antennas, so the OMP algorithm has an excessive demand on the number of antennas. In addition, thanks to the powerful learning of DNNs, the LAMP network and GM-LAMP network can converge faster than the AMP algorithm. Therefore, the complex multiplier required by the LAMP network and GM-LAMP network is smaller than that required by the AMP algorithm.

Next, we discuss the effects of resolution of ADC and channel sparsity on NMSE performance for two schemes based on DL. Figure 4 shows the performance changes of LAMP network schemes with different resolution ADC quantization. It can be seen that lower estimation error can be obtained by using higher-resolution ADC quantization, and the estimation performance after using mixed resolution ADC is better than four-bit ADC. With the increase in SNR, the performance difference between mixed resolution ADC and other low-resolution ADCs increases.

Figure 5 plots the variation of estimation error in low-resolution ADCs from one to four bits and mixed resolution after adopting the GAMP network scheme. The sparsity in this figure is equivalent to the number of channel paths ($L_{k}$). With the increase in sparsity, the channel estimation performance of all quantization resolutions will decline. It can be seen from the four curves in the comparison that the higher the resolution, the smaller the estimation error.

In addition, we evaluate the effects of NMSE and SNR on the total rate of beam selection in beamspace channel estimation. In this simulation, we reference the parameters of [37] to model the estimated beamspace channel $\hat{H}$ in case of imperfect CSI as
(28)$$\hat{H}=\alpha \tilde{H}+\sqrt{1-\alpha }E$$
where α∈(0,1) is the error parameter, E denotes the error matrix, whose elements satisfies the independent and identical distribution of the zero means and variance error, i.e., CN(0, NMSE), and $\tilde{H}=\left[{\tilde{h}}_{1},{\tilde{h}}_{2},\cdots ,{\tilde{h}}_{K}\right]$ denotes the channel matrix with perfect CSI for *K* users. In order to demonstrate the effectiveness of this work, we use the results provided in [37]; that is, AI beam selection with perfect CSI as the benchmark.

In Figure 6, we consider UPA based on the Saleh-Valenzuela channel model, and compare the sum-rate between an imperfect CSI and a perfect CSI with a downlink SNR of 10 dB. As shown in Figure 6, for the imperfect CSI under the mixed-resolution ADC architecture, when the NMSE is about −23 dB, the total rate-loss due to beam selection is less than 5% compared to the perfect CSI, and the beam-selection rate quantized by a one-bit ADC is always quite different from that of the mixed-resolution ADC. Figure 7 shows the effect of the change of SNR on the total rate of beam selection for an imperfect CSI and a perfect CSI when the NMSE is set to −10 dB. We can clearly see that with the increase in SNR, the beam selection rate of all three increases, but the gap between the rate of imperfect CSI and perfect CSI gradually increases.

## 6. Conclusions

In this paper, the channel estimation for a mmWave massive MIMO system with a mixed-resolution ADC architecture was investigated, where the BS deployed the lens antenna arrays. By using the sparsity of the mmWave channel, the beamspace channel estimation can be expressed as a sparse signal-recovery problem, and the channel can be recovered by the algorithm based on compressed sensing. We compare the traditional channel-estimation scheme with the channel-estimation scheme using DL. Simulation results showed that the performance of the DNN-based estimation scheme is significantly better than that of the traditional estimation scheme in the same configuration. Furthermore, the performance gap between using mixed-resolution ADC and using high-resolution ADC is slowly closing as the SNR increases. In future work, we will investigate the performance of IRS-assisted mmWave massive MIMO systems with mixed-ADC architectures, which aims to study the channel estimation problems in cascaded channels.

## Figures and Tables

**Figure 1 sensors-22-03938-f001:**
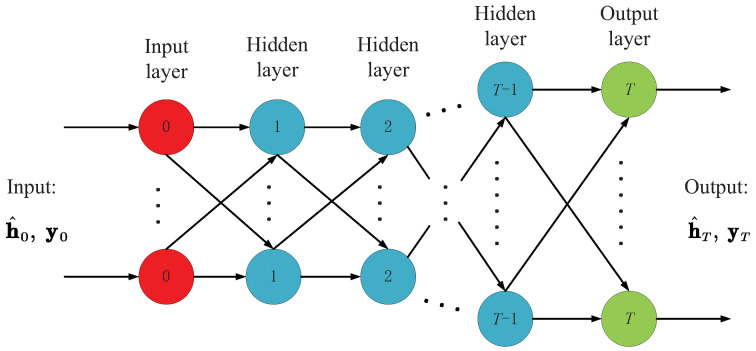
LAMP network structure with a total of *T* layers.

**Figure 2 sensors-22-03938-f002:**
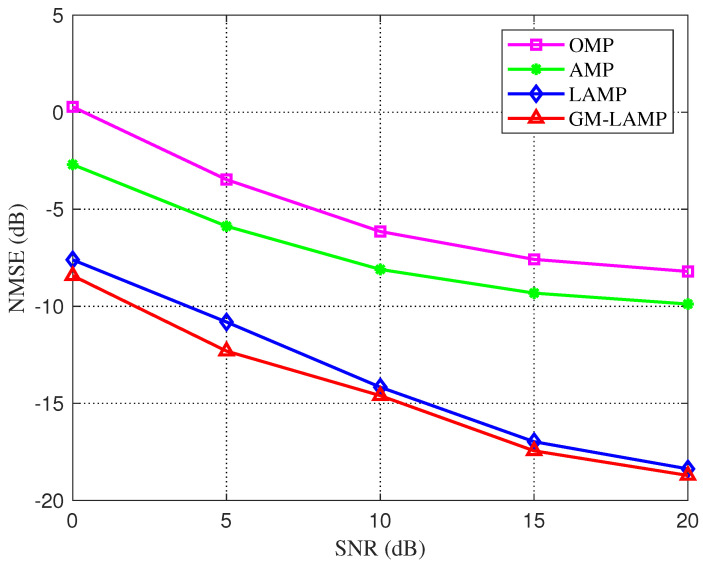
NMSE performance comparison of each channel estimation scheme for UPAs based on the Saleh-Valenzuela channel model in the mixed resolution ADC architecture.

**Figure 3 sensors-22-03938-f003:**
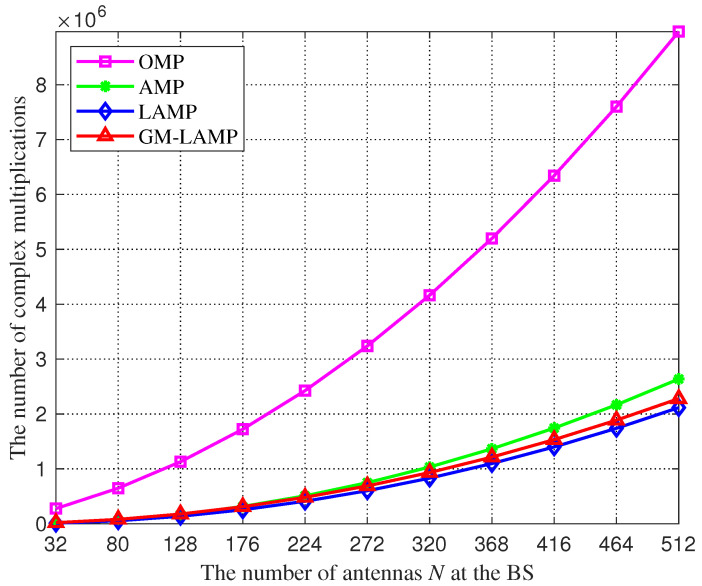
The number of complex multiplications against the number of antennas *N*.

**Figure 4 sensors-22-03938-f004:**
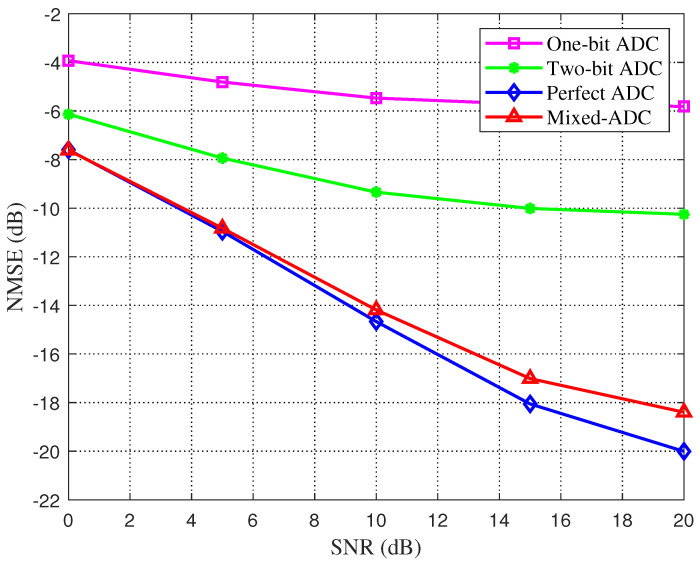
Channel estimation error is a function of SNR. Variations of estimation errors in different resolution ADCs quantification in LAMP network schemes are plotted.

**Figure 5 sensors-22-03938-f005:**
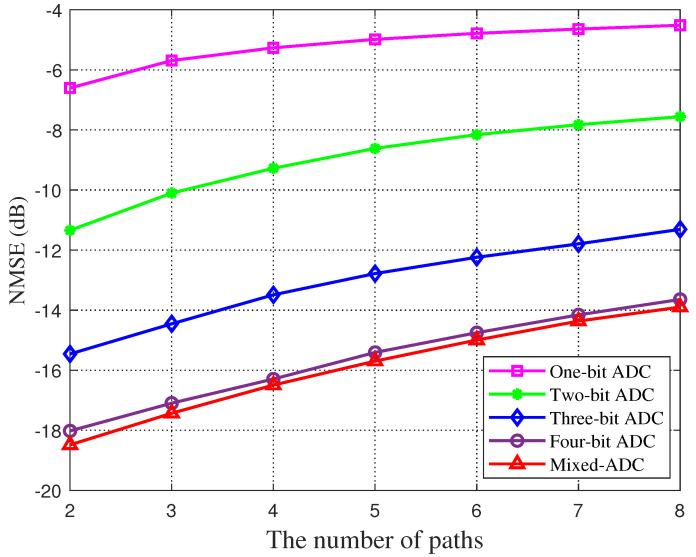
Channel estimation error is a function of channel sparsity. Variations of estimation errors in different resolution ADCs quantification in GAMP network schemes are plotted.

**Figure 6 sensors-22-03938-f006:**
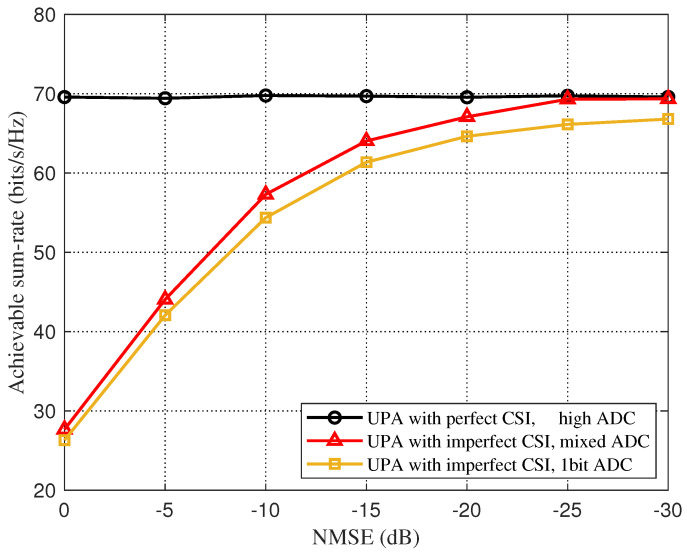
Total rate for beam selection against different NMSE for the beamspace channel estimation.

**Figure 7 sensors-22-03938-f007:**
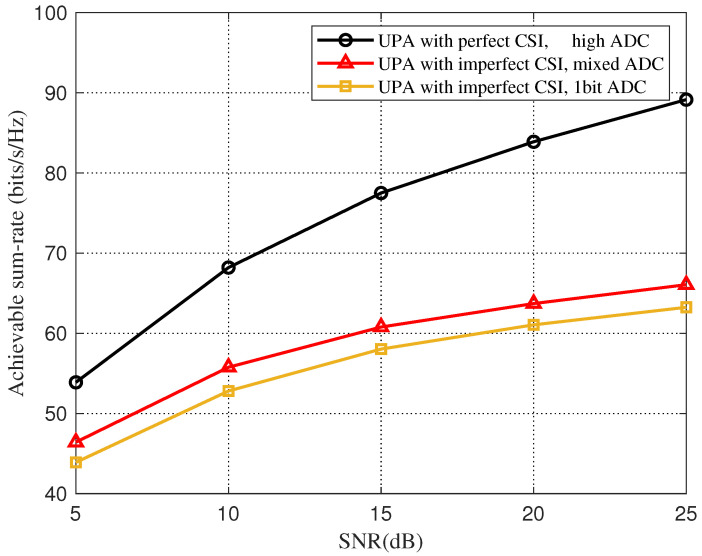
Total rate for beam selection against different SNR for the beamspace channel estimation.

**Table 1 sensors-22-03938-t001:** Multiple solutions to the problem of overwhelming hardware costs and power consumption in mmWave massive MIMO system.

Solutions	Advantages	Limitations
Lens antenna arrays	Significant reduction in the number of RF chains by adopting a hybrid precoding scheme.	Beam selection requires the BS to obtain the CSI of beamspace, and system performance is vulnerable to bandwidth.
Low-resolution ADCs	Reduce power consumption by reducing the high resolution of the ADCs to low resolution.	Low-resolution ADCs are prone to severe non-linear distortion.

## Data Availability

Not applicable.

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
