# Peer review of "Deep Learning-Based Channel Estimation for mmWave Massive MIMO Systems in Mixed-ADC Architecture"

_sensors, 2022, doi:10.3390/s22103938_

Round 1

Reviewer 1 Report

The submitted manuscript is well written and the topic is very hot. The work seems novel and the relevant literature is well presented. The authors should improve the quality of the English (some minor grammatical errors are found in the text)

Author Response

We would like to thank the your and reviewers for handling the review of our manuscript and providing constructive comments on improving our ORIGINAL MANUSCRIPT Sensors-1713427. By taking into account all the valuable comments, we have revised this manuscript carefully. We hope that the responses to the comments address all the reviewers’ concerns and the revised manuscript is to your satisfaction. At the same time, we also have tried our best to enhance the presentation, and some flaws are corrected. For clarification, the comments are shown in italic font, our responses are shown in blue, and the corresponding revisions are shown in box in this response and also highlighted in red in the revised manuscript. In the following, we address all the comments and suggestions of reviewers and editor Point-by-Point. For convenience, the review comments of reviewers are copied below, in bold, and are followed by the corresponding responses.

We have completed the revision and responses according to the your comments, please find the latest revised paper and response letter in the attached files.

Reviewer 2 Report

Strengths:
 (+) The problem is well-defined.
 (+) The proposed method is well-explained.
 (+) The experiments are convincing.

Weaknesses:
 (-) There are English issues.
 (-) References are inadequate.
 (-) The introduction must be improved.
 (-) The related work section must be enhanced.
 (-) Some improvements are needed in the description of the method.

==== FORMAT ==== 

The title of the paper is too long. In general, it is not recommended to exceed ten words. 

==== ENGLISH ==== 

The paper has some typos and grammar errors. Therefore, the authors need to proofread the paper to eliminate all of them.

Some sentences are too long. Generally, writing short sentences with one idea per sentence is better.

==== FIGURES ==== 

The figures are too small. 
==== REFERENCES ==== 

The literature review is incomplete. Several relevant references are missing.The authors are strongly recommended to improve the number of references mentioned in their work.

==== INTRODUCTION ==== 

The authors should add more references in the introduction to support the claims.

The authors need to explain better the context of this research, including why the research problem is essential.

The introduction should clearly explain the fundamental limitations of prior work relevant to this paper.

Contributions should be highlighted more. It should be clear what is novel and how it addresses the limitations of prior work. 

==== RELATED WORK ==== 

The authors should clearly explain the differences between the prior work and the solution presented in this paper.

There should be a related work section.

The authors should add a table that compares the key characteristics of prior work to highlight their differences and limitations. The authors may also consider adding a line in the table to describe the proposed solution.

==== METHOD ==== 

It is essential to clearly explain what is new and what is not in the proposed solution. If some parts are identical, they should be appropriately cited, and differences should be highlighted.
The authors must add a figure depicting the DL model adopted in their work.

It is necessary to discuss the complexity of the proposed solution.

==== REPRODUCIBILITY ==== 

To ensure reproducibility of the results, the code of the proposed solution should be made publicly available on a website.

==== CONCLUSION ==== 

Some text must be added to discuss future work or research opportunities.

Author Response

(The authors gave the same response as above.)

Reviewer 3 Report

This paper presented the methametical modelling supported by simulation graphs. Though the work presented is mathematically driven, I could not see any novelty in this work. The authors need to revise the manuscript with clearly stating what is novel and how this work is different and better than others.

Author Response

(The authors gave the same response as above.)

Round 2

Reviewer 2 Report

Dear Authors, all comments I had were addressed. I'm satisfied with the current version of your work.